# NOT ALL INSTANCES ARE EQUALLY VALUABLE: TOWARDS INFLUENCE-WEIGHTED DATASET DISTILLATION

## ABSTRACT

Dataset distillation condenses large datasets into synthetic subsets, achieving performance comparable to training on the full dataset while substantially reducing storage and computation costs. Most existing dataset distillation methods assume that all real instances contribute equally to the process. In practice, real-world datasets contain both informative and redundant or even harmful instances, and directly distilling the full dataset without considering data quality can degrade model performance. In this work, we present **I**nfluence-**W**eighted **D**istillation (`IWD`), a principled framework that leverages influence functions to explicitly account for data quality in the distillation process. `IWD` assigns adaptive weights to each instance based on its estimated impact on the distillation objective, prioritizing beneficial data while downweighting less useful or harmful ones. Owing to its modular design, `IWD` can be seamlessly integrated into diverse dataset distillation frameworks. Our empirical results suggest that integrating `IWD` tends to improve the quality of distilled datasets and enhance model performance, with accuracy gains of up to 7.8%.

## 1 INTRODUCTION

High-quality training data are crucial for modern machine learning, directly affecting performance, robustness, and generalization. Yet, with global data volume projected to reach 163 zettabytes by 2025 (Reinsel et al., 2017), the exponential growth of training data imposes unprecedented demands on storage and computation, driving up both training time and cost. To address these challenges, dataset distillation (Wang et al., 2018; Yu et al., 2024; Zhao et al., 2020; Kim et al., 2022) has emerged as a powerful paradigm for distilling large training sets into synthetic subsets that can achieve high model performance while significantly reducing the memory footprint and training time of deep networks on large datasets. While effective, these methods have the following limitations.

**Limitations.** (1) Existing dataset distillation methods typically aggregate features from all real instances to distill the synthetic dataset. In practice, real-world datasets often contain redundant, or even harmful data instances that can degrade model quality, and treating these instances the same as others can reduce the effectiveness of the distilled set (Wang et al., 2025a). For example, outliers in real instances may introduce misleading gradients, causing the distilled data lowering generalization, while redundant instances, such as a large number of visually repetitive images, contribute little novel information and cause the synthetic dataset to underutilize its limited capacity. (2) Some methods (Sundar et al., 2023; Moser et al., 2024; Xu et al., 2024; Du et al., 2024) reduce the first limitation by pruning low-quality or uninformative instances based on loss values or data characteristics before distillation. However, simply discarding these instances may lead to substantial information loss, since low-quality data do not necessarily imply zero contribution to the distillation process, and overly aggressive pruning can remove potentially useful information.

**Observation.** Inspired by Koh & Liang (2017), we observe that influence functions can be leveraged to estimate the contribution of each real instance to the distillation process. Instances with higher influence scores typically provide more useful and complementary information that enhance the quality of the distilled dataset, whereas those with low or negative influence scores tend to be redundant or even harmful. As shown in Fig.1, the first row on the left illustrates redundant data,

where most instances belong to the same sub-category (airliners). Since airliners already account for a large proportion of the airplane class in `CIFAR10`, additional similar instances provide little new information and thus contribute less to the distilled dataset. The second row shows outliers, such as blurry images or those capturing only partial views of airplanes, which may introduce misleading features. The right part illustrates instances with higher influence scores, which are relatively rare such as bombers, helicopters, and fighters. These instances provide complementary information that enriches the distilled dataset and improve generalization. This insight motivates a weighting strategy that emphasizes influential instances while down-weighting less useful ones.

**Our Proposal.** Therefore, in this work, we propose **I**nfluence-**W**eighted **D**istillation (`IWD`), a principled and flexible framework that tightly integrates influence functions into the dataset distillation process. Specifically, we extend the classical influence function theory to the distillation setting and derive a novel formulation of influence under dataset distillation, which enables principled estimation of each real instance's contribution to the quality of the distilled dataset. Building on this formulation, `IWD` first computes an influence score for each training instance. We then use these influence scores as soft, adaptive weights, prioritizing data instances that are most beneficial to the distillation objective and downweighting those that are less informative or poten-

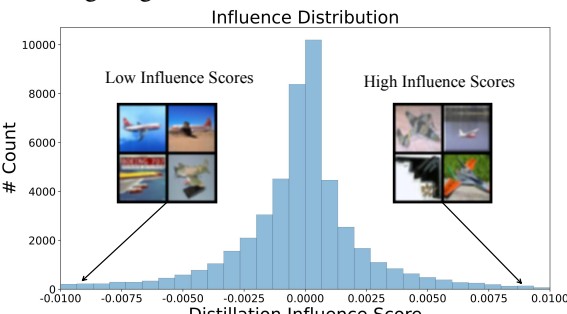

Figure 1: Distribution of instance influence scores in dataset distillation. Most scores concentrate around zero, reflecting that most real instances contribute only marginally. Low-influence instances often come from redundant sub-categories (e.g., airliners), whereas high-influence instances, though rare (e.g., bombers, fighters, helicopters), provide valuable information for distillation.

tially harmful. Importantly, `IWD` is designed as a modular plug-in that can be seamlessly incorporated into existing dataset distillation frameworks.

Our contributions are as follows:

(1) We extend classical influence function analysis to the dataset distillation setting, providing a new formulation for quantifying the contribution of each real data instance to the distillation process.

(2) We propose Influence-Weighted Distillation (`IWD`), a modular plug-in that incorporates influence-based data weighting into the distillation process. `IWD` is broadly compatible with existing dataset distillation frameworks and enhances the quality of synthetic datasets by prioritizing informative instances while downweighting less useful or harmful ones.

(3) We conduct extensive experiments on standard benchmarks to demonstrate that `IWD` consistently enhances the effectiveness of dataset distillation, achieving better generalization and model performance compared to previous approaches, with accuracy improvements of up to 7.8% (on `CIFAR10`).

## 2 RELATED WORK

**Dataset distillation**, also known as dataset condensation, aims to generate a synthetic dataset that can match the performance of training on the full dataset, while enabling significant savings in data storage and future retraining costs (Zhao et al., 2020; Cazenavette et al., 2022; Wang et al., 2018). Specifically, these works could be classified into 4 types: (1) *Meta-Model Matching* (Wang et al., 2018; Nguyen et al., 2021; Sucholutsky & Schonlau, 2021) optimizes the distilled data to maximize performance on real data. For instance, `DD` (Wang et al., 2018) employs a bi-level framework that updates the synthetic set by minimizing the loss on real data for models trained on it. (2) *Gradient Matching* (Zhao et al., 2020; Lee et al., 2022; Kim et al., 2022) optimizes the synthetic data so that their induced gradients resemble those of the original data, encouraging models trained on synthetic set to follow similar learning dynamics and achieve comparable performance. (3) *Trajectory Matching* (Cazenavette et al., 2022; Du et al., 2023) aligns the full optimization path of models trained on synthetic versus real data, minimizing trajectory discrepancies so that the distilled model mimics

the learning process of the real one. (4) *Distribution Matching* (Wang et al., 2025b; Zhao & Bilen, 2023) aligns feature distributions of synthetic and real data by using metrics such as MMD, enabling synthetic set to capture the underlying structure of the real dataset.

Despite technical differences, these strategies share the common objective of reducing the discrepancy between synthetic and real datasets in the matching process, while differing in the specific features they align, such as losses, gradients, trajectories, or distributions.

**Influence Function**, a fundamental concept in robust statistics, was originally introduced by (Koh & Liang, 2017) to characterize how an infinitesimal perturbation of a training instance propagates through the learning algorithm and impacts the model's parameters, predictions, and ultimately its performance. Subsequent research (Schioppa et al., 2021; Agarwal et al., 2017; Pruthi et al., 2020) extended influence functions to deep learning, where they have been employed for data valuation, error detection, domain adaptation, and related tasks.

## 3 PRELIMINARIES

### 3.1 DATASET DISTILLATION

Dataset distillation aims to condense the full dataset $D$ into a much smaller synthetic subset $S$, such that training a model ($\theta_S$) on $S$ yields performance comparable to, or even surpassing, the model ($\theta_D$) that achieved by training on the $D$ (Xu et al., 2024). Formally, given a large dataset $D = \{x_i, y_i\}_{i=1}^N$, the goal is to construct a synthetic set $S$ with $|S| \ll N$.

Specifically, DD (Wang et al., 2018) optimizes $S$ in a bi-level manner, where the inner loop trains a model on $S$ and the outer loop minimizes the loss of this model on the full dataset $D$.

$$\min_S \quad L(\theta_S; D) \quad \text{s.t.} \quad \theta_S = \arg\min_\theta L(\theta; S) \tag{1}$$

where $L$ is defined as: $L(\theta; D) = \frac{1}{N} \sum_{i=1}^N \ell(\theta; x_i, y_i)$, with $\ell$ denoting the loss function (e.g., cross-entropy).

DC (Zhao et al., 2020) seeks to construct a synthetic dataset $S$ whose training gradients closely match those induced by the original dataset $D$. In this way, the model updates based on $S$ can approximate the training trajectory on $D$.

$$\arg\min_S \mathbb{E}_{\theta_0 \sim P_{\theta_0}} \left[ \sum_{t=0}^T D(\nabla_\theta L^S(\theta^{(t)}), \nabla_\theta L^D(\theta^{(t)})) \right] \tag{2}$$

where $\theta_0 \sim P_{\theta_0}$ denotes initialization from a distribution (e.g., random initialization), $\theta^{(t)}$ denote the model parameters after $t$ training steps. $L^S$ and $L^D$ are the training losses on $S$ and $D$, respectively, and $D(\cdot, \cdot)$ is a distance metric (e.g., squared $\ell_2$ distance). In this formulation, DC updates $\theta^{(t)}$ by minimizing $L^S(\theta^{(t)})$ on the synthetic dataset. IDC (Kim et al., 2022) extends the DC that updates $\theta^{(t)}$ by minimizing the loss $L^D(\theta^{(t)})$ on the $D$. MTT (Cazenavette et al., 2022) improves upon DC by replacing single-step gradient alignment with trajectory-level matching over multiple iterations. MTT optimizes the synthetic dataset $S$ such that the parameter trajectory $\{\theta_S^{(t)}\}_{t=1}^N$ of the student model trained on $S$ closely follows the expert trajectory $\{\theta_D^{(t)}\}_{t=1}^M$ obtained from training on the full dataset $D$ (with $M \gg N$), thereby reducing the discrepancy between them.

Recent works (Moser et al., 2024; Sundar et al., 2023) extend the standard distillation framework by first pruning the original dataset $D$ to remove less informative or detrimental instances, resulting in a coreset $D_{core}$, and then distilling $D_{core}$ to obtain a representative synthetic set $S$.

### 3.2 CLASSICAL INFLUENCE FUNCTION

A fundamental question in data-centric machine learning is to quantify the impact of each training instance on the resulting model or a downstream evaluation metric. A natural and direct approach is to measure the change in model performance when a specific data instance is removed from the training set, known as the leave-one-out (LOO) effect. Formally, for a data instance $z = (x_j, y_j)$, the LOO effect on an evaluation metric $\mathcal{M}$ can be defined as:

$$\Delta \mathcal{M}(z) = \mathcal{M}(\theta_{D \setminus z}) - \mathcal{M}(\theta_D) \tag{3}$$

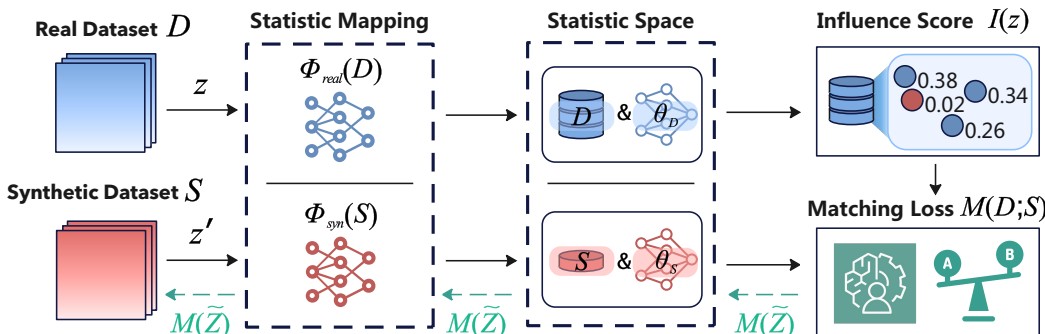

Figure 2: `IWD` Overview

where $\theta_{D \setminus z}$ and $\theta_D$ are the model parameters obtained by training on $D \setminus z$ and $D$, respectively.

However, computing the LOO effect exactly requires retraining the model $N$ times (once for each data instance), which is computationally expensive for large datasets and deep models. To address this, the influence function (Koh & Liang, 2017) from robust statistics provides an efficient first-order approximation of the LOO effect. Specifically, consider an empirical risk minimization (ERM) objective over $D$:

$$\theta_D = \arg\min_\theta L(\theta; D) = \frac{1}{N} \sum_{i=1}^{N} \ell(\theta; x_i, y_i) \tag{4}$$

where $\ell(\cdot)$ is a per-instance loss (e.g., cross-entropy). Classic result (Ling et al., 1984) tells us that the influence of upweighting $z$ on the parameters $\theta$ is given by $\mathcal{I}_{\text{up,params}}(z) \overset{\text{def}}{=} \frac{d\theta^{\epsilon,z}}{d\epsilon}\big|_{\epsilon=0} = -H_\theta^{-1} \nabla_\theta L(z, \theta)$, where $H_\theta \overset{\text{def}}{=} \frac{1}{n} \sum_{i=1}^{n} \nabla_\theta^2 L(z_i, \theta)$ is the Hessian and is positive definite (PD) by assumption. Building on this parameter perturbation view, we can further ask how such a change in $\theta$ influences any differentiable evaluation metric $M$. By applying the chain rule, the influence function (Koh & Liang, 2017) gives:

$$\frac{d\mathcal{M}(z_{\text{test}}, \theta_{z,\epsilon})}{d\epsilon}\bigg|_{\epsilon=0} = -\nabla_\theta \mathcal{M}(\theta_D; z_{\text{test}})^\top H_\theta^{-1} \nabla_\theta \ell(\theta_D; z) \tag{5}$$

where $\mathcal{M}(\theta; z_{\text{test}})$ denotes a differentiable evaluation metric (such as test loss or an accuracy surrogate) evaluated on a test instance $z_{\text{test}}$. This influence score captures how a marginal upweighting of a training instance $z$ propagates through the learned parameters to affect the chosen evaluation metric $\mathcal{M}$, and has been widely applied in data valuation, sample reweighting, and dataset pruning .

## 4 METHODOLOGY

While such "Prune then distill" approaches (Moser et al., 2024; Sundar et al., 2023) can enhance generalization, they may also discard valuable information from the full dataset, potentially limiting the performance of the distilled model. To address these limitations, we propose **I**nfluence-**W**eighted **D**istillation (`IWD`), which reduces the influence of harmful data instances and emphasizes beneficial ones, thereby improving the quality of the distilled subset.

### 4.1 OVERVIEW

As shown in Fig. 2, we leverage influence functions to guide the dataset distillation process. Specifically, `IWD` introduces a distillation-specific formulation that quantifies how the contribution of a real instance $z \in D$ propagates through the construction of the synthetic dataset $S$ and ultimately impacts the distillation objective $\mathcal{M}(S; D)$. Based on this formulation, `IWD` computes an influence score for each real instance by tracing its effect on $S$, and these scores are then used as adaptive weights—highlighting instances with positive influence while suppressing redundant or harmful ones.

### 4.2 INFLUENCE FUNCTION FOR DATASET DISTILLATION

**Distillation Objective Formulation.** We cast dataset distillation as minimizing a discrepancy between *matchable statistics* of the $S$ and $D$, evaluated along a reference training trajectory:

$$\min_S \ \mathbb{E}_{\theta_0 \sim P_{\theta_0}} \sum_{t=1}^{|T|} \mathcal{D}\Big(\Phi_{syn}(S; \theta_t), \ \Phi_{real}(D; \theta_t)\Big) \quad \text{s.t.} \quad \theta_t = \mathcal{U}_t\big(\theta_0; S_{\text{inner}}\big) \tag{6}$$

Here, $\mathcal{U}_t$ denotes the training dynamics up to step $t$ starting from $\theta_0$, $S_{\text{inner}} \in \{S, D\}$ specifies the dataset used for the inner training, $\Phi_{syn}$ and $\Phi_{real}$ extract *matchable statistics* from $S$ and $D$ under parameters $\theta_t$ (e.g., predictions, gradients, features, or trajectory states), and $\mathcal{D}(\cdot, \cdot)$ is a chosen discrepancy (e.g., supervised loss, $\ell_2$ distance, MMD, or an adversarial loss).

**Perturbing a single real instance.** We analyze the effect of perturbing a single real instance $z_j \in D$ on the distillation objective $\mathcal{M}(S; D)$. Let $D = \{z_i\}_{i=1}^N$ carry weights $w \in \Delta^{N-1}$, and upweight $z_j$ by $\varepsilon$: $w^\varepsilon = w + \varepsilon \, \mathbf{e}_j$, $D^\varepsilon := \{(z_i, w_i^\varepsilon)\}_{i=1}^N$. This induces perturbed real statistics $\Phi_{real}^\varepsilon(D; \theta_t) := \Phi_{real}(D^\varepsilon; \theta_t)$. The perturbed distillation objective is then

$$\mathcal{M}_\varepsilon(S; D) := \mathbb{E}_{\theta_0 \sim P_{\theta_0}} \sum_{t=1}^{|T|} \mathcal{D}\Big(\Phi_{syn}(S; \theta_t), \ \Phi_{real}^\varepsilon(D; \theta_t)\Big), \tag{7}$$

where the inner trajectory $\theta_t = \mathcal{U}_t(\theta_0; S_{\text{inner}})$ is determined by the chosen inner-training dataset $S_{\text{inner}}$. If $S_{\text{inner}} = S$, the perturbation $\varepsilon$ influences only the real statistics $\Phi_{real}$, thereby altering the distillation objective $\mathcal{M}(S; D)$ while leaving $\theta_t$ unaffected. In contrast, when $S_{\text{inner}} = D$, the perturbation propagates into both the real statistics $\Phi_{real}$ and the inner trajectory $\theta_t$.

**Definition (Distillation Influence Function).** For a fixed choice of the inner-training dataset $S_{\text{inner}} \in \{S, D\}$, the influence of upweighting $z_j \in D$ by $\varepsilon$ on $\mathcal{M}(S; D)$ is

$$\mathcal{I}(z_j; S) = \frac{d}{d\varepsilon} \mathcal{M}_\varepsilon(S; D)\Big|_{\varepsilon=0} = \mathbb{E}_{\theta_0 \sim P_{\theta_0}} \sum_{t=1}^{|T|} \frac{d}{d\varepsilon} \mathcal{D}(a_t^\varepsilon, b_t^\varepsilon)\Big|_{\varepsilon=0} \tag{8}$$

where, for brevity, $a_t := \Phi_{syn}(S; \theta_t)$, $b_t := \Phi_{real}(D; \theta_t)$. We then denote $\nabla_1 \mathcal{D}(a_t, b_t)$ and $\nabla_2 \mathcal{D}(a_t, b_t)$ be the gradients of $\mathcal{D}$ (its first and second argument respectively). Then, we get:

$$\frac{d}{d\varepsilon} \mathcal{D}(a_t^\varepsilon, b_t^\varepsilon) = \Big\langle \nabla_1 \mathcal{D}(a_t, b_t), \frac{d}{d\varepsilon} a_t^\varepsilon \Big\rangle + \Big\langle \nabla_2 \mathcal{D}(a_t, b_t), \frac{d}{d\varepsilon} b_t^\varepsilon \Big\rangle \tag{9}$$

By the chain rule through the trajectory,

$$\frac{d}{d\varepsilon} a_t^\varepsilon = J_t^{syn} \underbrace{\frac{d}{d\varepsilon} \theta_t^\varepsilon}_{=: \, u_{t,j}^\varepsilon}, \quad \frac{d}{d\varepsilon} b_t^\varepsilon = \underbrace{\frac{\partial}{\partial\varepsilon} \Phi_{real}(D^\varepsilon; \theta)\Big|_{\theta=\theta_t^\varepsilon}}_{=: \, s_{t,j}^{\text{real},\varepsilon}} + J_t^{real} \underbrace{\frac{d}{d\varepsilon} \theta_t^\varepsilon}_{u_{t,j}^\varepsilon} \tag{10}$$

where, for brevity, $J_t^{syn} := \partial_\theta \Phi_{syn}(S; \theta_t)$, $J_t^{real} := \partial_\theta \Phi_{real}(D; \theta_t)$. Evaluating at $\varepsilon = 0$ yields the shorthands $u_{t,j} := u_{t,j}^0$, $s_{t,j}^{\text{real}} := s_{t,j}^{\text{real},0}$, $a_t := a_t^0$, $b_t := b_t^0$.

Substituting into (9) at $\varepsilon = 0$ gives

$$\frac{d}{d\varepsilon} \mathcal{D}(a_t^\varepsilon, b_t^\varepsilon)\Big|_{\varepsilon=0} = \big\langle \nabla_1 \mathcal{D}(a_t, b_t), \ J_t^{\text{syn}} u_{t,j} \big\rangle + \big\langle \nabla_2 \mathcal{D}(a_t, b_t), \ s_{t,j}^{\text{real}} + J_t^{\text{real}} u_{t,j} \big\rangle. \tag{11}$$

Combining (8) and (11), we obtain the *general decomposition*

$$\mathcal{I}(z_j; S) = \mathbb{E}_{\theta_0} \sum_{t=1}^{|T|} \Big[ \underbrace{\big\langle \nabla_2 \mathcal{D}(a_t, b_t), \ s_{t,j}^{\text{real}} \big\rangle}_{\text{explicit (real-stats)}} + \underbrace{\big\langle J_t^{\text{syn}\top} \nabla_1 \mathcal{D}(a_t, b_t) + J_t^{\text{real}\top} \nabla_2 \mathcal{D}(a_t, b_t), \ u_{t,j} \big\rangle}_{\text{implicit via trajectory}} \Big]$$

---

**Algorithm 1:** Influence-Weighted Dataset Distillation (IWD)

---

**Input:** Dataset $D$; initialization distribution $p(\theta_0)$; number of distilled samples $M$; steps $T$;
       batch size $n$; initial learning rate $\eta_0$; softmax temperature $\tau$
**Output:** Weighted distilled data $\tilde{\mathbf{z}}$, learning rate $\tilde{\eta}$
**Initialization:** $\tilde{\mathbf{z}} = \{\tilde{z}_i\}_{i=1}^M, \quad \tilde{\eta} \leftarrow \eta_0$
     // Influence-Weighted Distillation
**1 for** $t = 1$ **to** $T$ **do**
**2**     Sample minibatch $\mathbf{z}_t$ of $n$ real samples from $D$
**3**     Sample initial model weights $\theta_0$ from $p(\theta_0)$
**4**     Update model parameters using distilled data
        // Influence Score Estimation
**5**     **for** *each real sample $z \in D$* **do**
**6**       Estimate influence score $\mathcal{I}_t(z)$ based on current $\tilde{\mathbf{z}}$
**7**     $w_t(z) \leftarrow \text{softmax}(\mathcal{I}_t(z)/\tau)$
**8**     Compute weighted loss on $\mathbf{z}_t$:
**9**     $\mathcal{L}_{\text{total}} = \sum_{z \in \mathbf{z}_t} w_t(z) \cdot \mathcal{M}(\tilde{\mathbf{z}}, z)$
**10**    Update $\tilde{\mathbf{z}}$ and $\tilde{\eta}$ via gradient descent on $\mathcal{L}_{\text{total}}$

---

The decomposition shows that the influence score naturally consists of an *explicit* term, capturing the direct contribution of $z_j$ to the real-data statistics, and an *implicit* term, propagating through the parameter trajectory via $u_{t,j}$. The form of $u_{t,j}$ depends on the choice of the inner-training set $S_{\text{inner}}$:

If $S_{\text{inner}} = S$, then $z_j \notin S_{\text{inner}}$ and the parameters are not updated with respect to $z_j$. Hence $u_{t,j} = 0$, and only the explicit term remains.

If $S_{\text{inner}} = D$, then the parameters are trained on the real dataset, and $u_{t,j}$ follows the classical influence function $characterization$ ($Lin$ $et\ al.$, $1984$), $u_{t,j} = \left. \frac{d\theta_t^\varepsilon}{d\varepsilon} \right|_{\varepsilon=0} = -H_{\theta_t}^{-1} \nabla_\theta \ell(\theta_t; z_j)$, where $H_{\theta_t}$ is the Hessian of the training loss at $\theta_t$. In this case, the implicit term dominates, and the formulation recovers the standard influence-function perspective.

### 4.3 ALGORITHM WORKFLOW

Given a real dataset $D$, the proposed influence-weighted dataset distillation algorithm proceeds iteratively as outlined in Algorithm 1. At each iteration $t$, a minibatch $\mathbf{z}_t$ of $n$ real samples is drawn from $D$, and the model is initialized with $\theta_0 \sim p(\theta_0)$. The model is first updated using the current synthetic dataset $\tilde{\mathbf{z}}$ (Lines 2–4).

Next, the influence score $\mathcal{I}_t(z)$ of each real instance $z \in D$ is estimated with respect to the current synthetic dataset (Lines 5–6). These scores are normalized through a softmax function with temperature $\tau$, yielding per-instance weights $w_t(z)$. The influence-weighted distillation loss is then computed over the minibatch $\mathbf{z}_t$ as the weighted sum of matching losses (Lines 7–9).

Finally, both the synthetic dataset $\tilde{\mathbf{z}}$ and the learning rate $\tilde{\eta}$ are updated via gradient descent on this total loss (Line 10). This procedure is repeated for $T$ iterations, gradually shaping $\tilde{\mathbf{z}}$ to emphasize highly influential instances while mitigating the impact of harmful ones.

## 5 EXPERIMENTS

In this section, we empirically evaluate the effectiveness of the proposed Influence-Weighted Distillation (`IWD`) framework on standard dataset distillation benchmarks. We examine whether `IWD` can improve over state-of-the-art baselines, assess the benefits of influence-guided weighting relative to uniform weighting methods, and evaluate its robustness across different datasets and architectures.

**Datasets.** All experiments are conducted on widely used benchmark datasets for dataset distillation, including `CIFAR10` (Krizhevsky et al., 2009) (60,000 $32 \times 32$ color images in 10 classes),

| Method | CIFAR10 | | | CIFAR100 | | | SVHN | | |
|---|---|---|---|---|---|---|---|---|---|
| | 1 | 10 | 50 | 1 | 10 | 50 | 1 | 10 | 50 |
| Random | $14.4_{\pm2.0}$ | $26.0_{\pm1.2}$ | $43.4_{\pm1.0}$ | $4.2_{\pm0.3}$ | $14.6_{\pm0.5}$ | $30.0_{\pm0.4}$ | $14.6_{\pm1.6}$ | $35.1_{\pm4.1}$ | $70.9_{\pm0.9}$ |
| Herding | $21.5_{\pm1.2}$ | $31.6_{\pm0.7}$ | $40.4_{\pm0.6}$ | $8.4_{\pm0.3}$ | $17.3_{\pm0.3}$ | $33.7_{\pm0.5}$ | $20.9_{\pm1.3}$ | $50.5_{\pm3.3}$ | $72.6_{\pm0.8}$ |
| DC | $28.3_{\pm0.5}$ | $44.9_{\pm0.5}$ | $53.9_{\pm0.5}$ | $12.8_{\pm0.3}$ | $25.2_{\pm0.3}$ | $32.1_{\pm0.3}$ | $31.2_{\pm1.4}$ | $76.1_{\pm0.6}$ | $82.3_{\pm0.3}$ |
| IWD+DC | $29.6_{\pm0.6}$ | $52.7_{\pm0.6}$ | $61.0_{\pm0.4}$ | $14.2_{\pm0.5}$ | $32.4_{\pm0.2}$ | $36.5_{\pm0.3}$ | $34.4_{\pm1.4}$ | $78.6_{\pm0.1}$ | $83.4_{\pm0.3}$ |
| IDM | $45.6_{\pm0.7}$ | $58.6_{\pm0.1}$ | $67.5_{\pm0.5}$ | $20.1_{\pm0.3}$ | $45.1_{\pm0.1}$ | $50.0_{\pm0.2}$ | — | — | — |
| KIP | $49.9_{\pm0.2}$ | $62.7_{\pm0.3}$ | $68.6_{\pm0.2}$ | $15.7_{\pm0.2}$ | $28.3_{\pm0.1}$ | — | $57.3_{\pm0.1}$ | $75.0_{\pm0.1}$ | $80.5_{\pm0.1}$ |
| MTT | $46.3_{\pm0.8}$ | $65.6_{\pm0.7}$ | $71.6_{\pm0.2}$ | $24.3_{\pm0.3}$ | $40.1_{\pm0.4}$ | $47.7_{\pm0.2}$ | — | — | — |
| FRePo | $46.8_{\pm0.7}$ | $65.5_{\pm0.4}$ | $71.7_{\pm0.2}$ | $\underline{28.7}_{\pm0.1}$ | $42.5_{\pm0.2}$ | $44.3_{\pm0.2}$ | — | — | — |
| IDC | $50.6_{\pm0.4}$ | $67.5_{\pm0.5}$ | $74.5_{\pm0.1}$ | $28.4_{\pm0.1}$ | $45.1_{\pm0.3}$ | — | $68.5_{\pm0.9}$ | $87.5_{\pm0.3}$ | $90.1_{\pm0.1}$ |
| IWD+IDC | $\mathbf{51.5}_{\pm0.5}$ | $68.3_{\pm0.6}$ | $74.9_{\pm0.3}$ | $\mathbf{29.1}_{\pm0.3}$ | $46.1_{\pm0.5}$ | — | $\mathbf{70.1}_{\pm0.3}$ | $\mathbf{88.2}_{\pm0.4}$ | $\mathbf{90.8}_{\pm0.1}$ |
| PDD | — | $67.9_{\pm0.2}$ | $76.5_{\pm0.4}$ | — | $45.8_{\pm0.5}$ | $53.1_{\pm0.4}$ | — | — | — |
| IWD+PDD | — | $\mathbf{68.8}_{\pm0.2}$ | $\mathbf{76.9}_{\pm0.3}$ | — | $\mathbf{46.7}_{\pm0.4}$ | $\mathbf{53.5}_{\pm0.3}$ | — | — | — |
| Full Dataset | | $84.8_{\pm0.1}$ | | | $56.2_{\pm0.3}$ | | | $95.4_{\pm0.1}$ | |

Table 1: Dataset distillation performance of state-of-the-art and the proposed **IWD**. The best results are highlighted in **bold**, the second best are underlined, and methods with a gray background correspond to variants augmented with IWD.

CIFAR100 (Krizhevsky et al., 2009) (60,000 $32 \times 32$ images in 100 classes) and SVHN (Netzer et al., 2011) (over 99,000 $32 \times 32$ digit images in 10 classes).

**Baselines.** We consider both data selection and distillation algorithms as baselines, including Random, Herding(Welling & Bren) for selection and DC (Zhao et al., 2020), KIP (Nguyen et al., 2021), MTT (Cazenavette et al., 2022), FRePo (Zhou et al., 2022), IDC (Kim et al., 2022), RFAD-NN (Loo et al., 2022), IDM (Zhao et al., 2023), DREAM (Liu et al., 2023) and PDD (Chen et al., 2023) for distillation. Herding selects real instances closest to the class mean in feature space to best represent the real data distribution. All baseline results in the main experiments are taken from the current state-of-the-art reports in the respective original papers or public benchmarks to ensure a fair and up-to-date comparison.

**Architectures and Distillation Settings.** We adopt the original configurations of each baseline, with modifications only to incorporate our weighted loss. Our framework is applied to three representative dataset distillation methods: DC, IDC and PDD, where we use the optimal hyper-parameters reported in their original papers. To further assess the transferability of the distilled data, we evaluate models trained on the synthetic datasets using CovNet-3 and ResNet-10 (He et al., 2015).

**Evaluation.** Once the synthetic subsets have been constructed for each dataset, they are used to train randomly initialized networks from scratch, followed by evaluation on their corresponding testing sets. For each experiment, we report the mean and the standard deviation of the testing accuracy of 5 trained networks. To train networks from scratch at evaluation time, we use the SGD optimizer with a momentum of 0.9 and a weight decay of $5 \times 10^{-4}$. For DC, the learning rate is set to be 0.1. For IDC, the learning rate is set to be 0.01. For PDD, the learning rate is set to be 0.01. During the evaluation time, we follow the augmentation methods of each method to train networks from scratch.

## 5.1 DISTILLATION RESULTS

**Overall Performance.** Table 1 presents the test accuracy of our proposed IWD framework, as well as all baselines, on CIFAR10, CIFAR100, and SVHN under varying numbers of distilled images per class (1, 10, 50). Across all settings, IWD consistently outperforms its base counterparts.

We see that all combinations—IWD+DC, IWD+IDC and IWD+PDD—consistently yield large improvements over their respective base distillation frameworks across all datasets and IPCs. Specifically, IWD + DC outperforms DC by significant margins of 1.3%/7.8%/6.1% on CIFAR10, 1.4%/7.2%/4.4% on CIFAR100, and 3.2%/2.5%/1.1% on SVHN. IWD + IDC outperforms IDC by significant margins of 0.9%/0.8%/0.4% on CIFAR10, 0.7%/1.0% on CIFAR100($IPC = 1/10$), and 1.6%/0.7%/0.7% on SVHN. IWD + PDD outperforms PDD by significant margins of 0.9%/0.4% on CIFAR10($IPC = 10/50$) and 0.9%/0.4% on CIFAR100($IPC = 10/50$).

Compared with their uniform-weighting counterparts, the improvements are substantial, suggesting that `IWD` effectively mitigates the limitations of uniform weighting. Moreover, for `DC`, `IDC` and `PDD`, which already incorporate more sophisticated objectives, `IWD` still yields additional gains, demonstrating the general applicability of `IWD` across diverse distillation frameworks.

## 5.2 ABLATION STUDY

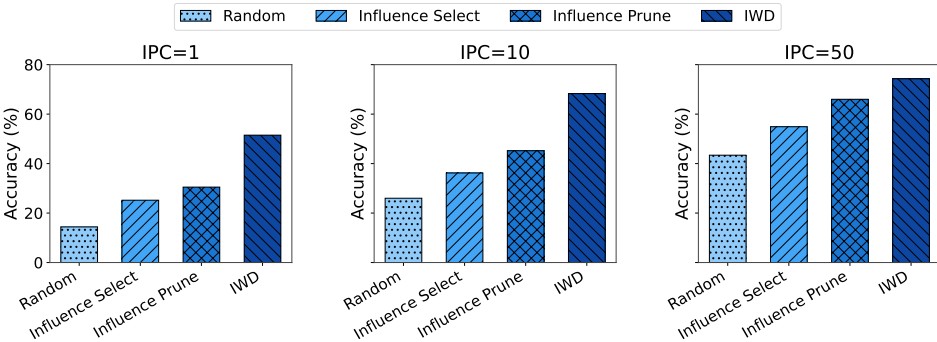

Figure 3: Ablation Study of Influence Weighting. Results on `CIFAR10` ($IPC = 1, 10, 50$) show that `IWD` consistently outperforms `Random-Select`, `Influence-Select`, and `Influence-Prune`, demonstrating the effectiveness of weighting instances by influence.

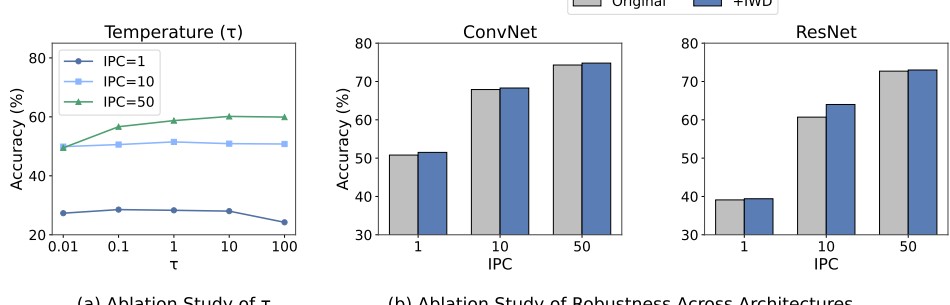

(a) Ablation Study of τ    (b) Ablation Study of Robustness Across Architectures.

Figure 4: Ablation studies on `CIFAR10`. (a) Ablation Study of $\tau$. Effect of softmax temperature $\tau$ under varying IPCs, showing a unimodal accuracy trend where moderate $\tau$ achieves the best balance between emphasizing high-influence instances and retaining global information. (b) Ablation Study of Architectures. Robustness across architectures (`ConvNet-3`, `ResNet-10`), where `IWD` consistently outperforms corresponding baselines, demonstrating generality beyond specific backbones.

**Ablation Study of Influence Weighting.** We compare four methods to understand how instance influence should be exploited during distillation: (1) *Random-Select*: uniformly sample real instances and train the model directly on this subset. (2) *Influence-Select*: use only the most influential real instances and train the model directly on this subset. (3) *Influence-Prune-then-Distill*: remove the lowest-influence tail of the real data and run the standard distillation algorithm on the remaining set (we retain the highest-influence 90% by default). (4) `IWD` (ours): use all real instances but weight them by the softmax of their influence scores (Sec. 4), thereby emphasizing helpful instances while down-weighting harmful ones during distillation. We conduct this ablation on `CIFAR10` under varying IPCs (1, 10, 50), following the same experimental setup as in Sec. 5.

Across datasets and IPC settings, we observe a consistent ordering: *Random-Select* < *Influence-Select* < *Influence-Prune-then-Distill* < `IWD`. *Influence-Select* is better than *Random-Select* because it focuses training on the most helpful instances. *Influence-Prune-then-Distill* further improves upon *Influence-Select* by still retaining a broader pool of real data for distillation, while discarding the clearly harmful instances. `IWD` achieves the best results since it adaptively reweights all instances instead of making hard selections: influential samples receive higher emphasis, harmful ones are

down-weighted rather than discarded, and the information of the full dataset is preserved for the distillation process.

**Ablation Study of $\tau$.** We further investigate the effect of the softmax temperature $\tau$ in Alg. 1. Experiments are conducted on `CIFAR10` under varying IPCs, where $\tau$ is chosen from $0.01, 0.1, 1, 10, 100$. As shown in Fig.4, we observe a clear unimodal behavior: for a fixed IPC, test accuracy first increases with $\tau$, reaches a peak, and then declines as $\tau$ continues to grow. This is because small values of $\tau$ overemphasizes a few high-influence instances and overlook the global information, while very large values flatten the distribution, approaching uniform weighting and diminishing the benefit of influence guidance. Moderate values of $\tau$ strike a balance by highlighting helpful instances while still allowing sufficient global information to contribute. Moreover, we observe that the optimal $\tau$ increases with larger $IPC$. This is because as $IPC$ grows, more distilled images are generated, while the information carried by each individual image is limited. With more distilled images, a larger number of real instances can contribute to the distillation process, many of which provide useful information. Consequently, a larger $\tau$ is required to soften the weighting distribution, allowing more instances to be effectively utilized rather than relying only on a few dominant ones.

**Ablation Study of Architectures.** To further examine the robustness and generalizability of the proposed `IWD` framework, we conduct experiments on multiple neural network architectures, including a lightweight `ConvNet-3` and a deeper `ResNet-10`. For each architecture, we apply both our influence-weighted distillation and the corresponding baseline methods, and evaluate the resulting test accuracies on `CIFAR10`.

As illustrated in Figure 4, `IWD` consistently yields higher accuracy than the baselines across all tested architectures. These improvements remain stable despite substantial differences in network depth and representational capacity, highlighting that the effectiveness of `IWD` is not tied to a specific backbone design but rather provides a generally applicable enhancement to dataset distillation.

**Synthesized Samples Visualization.** In Fig. 5, we show synthesized samples on `CIFAR10` distilled by `IWD+DC` at $IPC = 1$, sorted by their estimated influence scores. We find that **low-influence** samples usually fall into two categories: (i) very simple and easy images that are almost trivial to recognize, thus offering little useful gradient signal; or (ii) redundant or ambiguous images whose gradients are unstable or even misleading. On the other hand, **high-influence** samples tend to be (i) images containing rich class-specific features such as distinctive shapes, edges, or colors; or (ii) harder but still informative cases, like unusual viewpoints or partial occlusions, that provide strong learning signals. This aligns with the goal of `IWD`: the weighting mechanism emphasizes samples that supply valuable global information—either prototypical or challenging—while down-weighting overly easy or redundant ones.

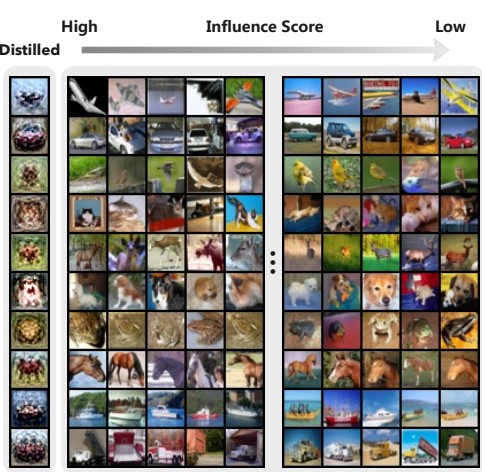

Figure 5: Synthesized images of `CIFAR10` using `IWD+ DC`.

## 6 CONCLUSION

We presented **Influence-Weighted Distillation (`IWD`)**, a framework that integrates influence functions into dataset distillation. By assigning adaptive weights to real instances, `IWD` prioritizes beneficial data while mitigating the impact of harmful ones, and can be seamlessly applied to existing distillation methods. Experiments on standard benchmarks show that `IWD` consistently improves performance across datasets, IPCs, and architectures. These results demonstrate that influence-guided weighting is a simple yet effective strategy to enhance dataset distillation. Our code is available at https://anonymous.4open.science/r/Influence-Weighted-Distillation-4DE1.

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
