## A  STATEMENTS

### A.1  REPRODUCIBILITY STATEMENT

We are committed to ensuring the reproducibility of our results. To this end, we provide the complete source code, training scripts, and instructions for reproducing all experiments in an anonymous GitHub repository.

### A.2  LLMS USAGE STATEMENT

Large Language Models (LLMs) were used solely as general-purpose assistive tools in the preparation of this work. Specifically, we employed LLMs to (i) polish the language of the manuscript for clarity and readability, (ii) verify parts of the mathematical derivations for correctness, and (iii) provide assistance with code implementation checks such as debugging and syntax refinement. All research ideas, methodological contributions, experimental designs, and analyses were conceived and executed entirely by the authors. The authors take full responsibility for all content presented in this paper.

**Example of LLM-assisted editing.**    Below we provide a concrete example in the caption of Fig.1 illustrating how an LLM was used to refine the clarity and conciseness of our writing.

>**Ours (original draft):**
>*Continue refining and editing the following pararaphs:*
>Most scores are concentrated around zero, since the majority of real data contribute only marginally to the construction of the synthetic set. Low-influence (potentially harmful) instances often originate from a narrow sub-category *thus supply redundant information* (e.g., airliners within airplanes), whereas high-influence (useful but relatively rare) instances tend to span more diverse sub-categories *thus supply rich information that the model does not ever see* (e.g., bombers, fighters, etc. within airplanes).
>
>**LLM's (revised draft):**
>Most scores are concentrated around zero, as the majority of real data contribute only marginally to the synthetic set. Low-influence (potentially harmful) instances often originate from a narrow sub-category, thereby providing redundant information (e.g., airliners within airplanes), whereas high-influence (useful but relatively rare) instances span more diverse sub-categories, thereby supplying novel and informative signals (e.g., bombers, fighters, etc. within airplanes).
>
>**Our final caption:**
>Most scores are concentrated around zero, as the majority of real data contribute only marginally to the synthetic set. Low-influence (potentially harmful) instances often originate from a narrow sub-category, thus providing redundant information (e.g., airliners within airplanes), whereas high-influence (useful but relatively rare) instances span more diverse sub-categories, thereby contributing diverse and valuable information (e.g., bombers, fighters, etc. within airplanes).

## B  APPENDIX

**Setup and notation.**    We consider the unified distillation objective

$$\mathcal{M}(S; D) := \mathbb{E}_{\theta_0 \sim P_{\theta_0}} \sum_{t=1}^{|T|} \mathcal{D}\Big(\Phi_{syn}(S; \theta_t), \ \Phi_{\mathrm{real}}(D; \theta_t)\Big), \quad \theta_t = \mathcal{U}_t\big(\theta_0; S_{\mathrm{inner}}\big), \qquad (12)$$

where $S_{\mathrm{inner}} \in \{S, D\}$ specifies the dataset used by the inner trajectory.

To upweight a real instance $z_j \in D$, we use per-instance weights $w \in \Delta^{N-1}$.

And we define $w^{\varepsilon} = w + \varepsilon \, \mathbf{e}_j$, $D^{\varepsilon} := \{(z_i, w_i^{\varepsilon})\}_{i=1}^N$.

We write $\theta_t^\varepsilon := \mathcal{U}_t(\theta_0; S_{\text{inner}}^\varepsilon)$, where

$$S_{\text{inner}}^\varepsilon = \begin{cases} S, & \text{if } S_{\text{inner}} = S \quad \text{(inner uses synthetic data)}, \\ D^\varepsilon, & \text{if } S_{\text{inner}} = D \quad \text{(inner uses real data)}. \end{cases}$$

We also define the perturbed real statistics $\Phi_{\text{real}}^\varepsilon(D; \theta) := \Phi_{\text{real}}(D^\varepsilon; \theta)$.

**Definition (Distillation Influence Function).**   For a fixed choice of $S_{\text{inner}}$, we define

$$\mathcal{I}(z_j; S \mid S_{\text{inner}}) := \frac{d}{d\varepsilon} \mathcal{M}_\varepsilon(S; D) \Big|_{\varepsilon=0}, \tag{13}$$

$$\mathcal{M}_\varepsilon(S; D) := \mathbb{E}_{\theta_0} \sum_{t=1}^{|T|} \mathcal{D}\Big(\Phi_{syn}(S; \theta_t^\varepsilon), \ \Phi_{\text{real}}^\varepsilon(D; \theta_t^\varepsilon)\Big). \tag{14}$$

**Step 1: Differentiate the perturbed objective.**   Assuming regularity conditions that permit interchanging derivative and expectation/summation, we obtain

$$\mathcal{I}(z_j; S) = \mathbb{E}_{\theta_0} \sum_{t=1}^{|T|} \frac{d}{d\varepsilon} \mathcal{D}\Big(\Phi_{syn}(S; \theta_t^\varepsilon), \ \Phi_{\text{real}}^\varepsilon(D; \theta_t^\varepsilon)\Big) \Big|_{\varepsilon=0}. \tag{15}$$

**Step 2: Chain rule on the discrepancy.**   Let, for brevity,

$$a_t^\varepsilon := \Phi_{syn}(S; \theta_t^\varepsilon), \qquad b_t^\varepsilon := \Phi_{\text{real}}^\varepsilon(D; \theta_t^\varepsilon),$$

and denote by $\nabla_1 \mathcal{D}(a, b)$ and $\nabla_2 \mathcal{D}(a, b)$ the gradients of $\mathcal{D}$ w.r.t. its first and second argument, respectively. Then

$$\frac{d}{d\varepsilon} \mathcal{D}(a_t^\varepsilon, b_t^\varepsilon) = \Big\langle \nabla_1 \mathcal{D}(a_t^\varepsilon, b_t^\varepsilon), \ \frac{d}{d\varepsilon} a_t^\varepsilon \Big\rangle + \Big\langle \nabla_2 \mathcal{D}(a_t^\varepsilon, b_t^\varepsilon), \ \frac{d}{d\varepsilon} b_t^\varepsilon \Big\rangle. \tag{16}$$

**Step 3: Expand $\frac{d}{d\varepsilon} a_t^\varepsilon$ and $\frac{d}{d\varepsilon} b_t^\varepsilon$.**   By the chain rule through the trajectory,

$$\frac{d}{d\varepsilon} a_t^\varepsilon = \partial_\theta \Phi_{syn}(S; \theta_t^\varepsilon) \underbrace{\frac{d}{d\varepsilon} \theta_t^\varepsilon}_{=: u_{t,j}^\varepsilon}, \tag{17}$$

$$\frac{d}{d\varepsilon} b_t^\varepsilon = \underbrace{\frac{\partial}{\partial \varepsilon} \Phi_{\text{real}}(D^\varepsilon; \theta) \Big|_{\theta=\theta_t^\varepsilon}}_{=: s_{t,j}^{\text{real},\varepsilon}} + \partial_\theta \Phi_{\text{real}}(D^\varepsilon; \theta_t^\varepsilon) \underbrace{\frac{d}{d\varepsilon} \theta_t^\varepsilon}_{u_{t,j}^\varepsilon}. \tag{18}$$

We introduce the Jacobians

$$J_t^{syn}(\varepsilon) := \partial_\theta \Phi_{syn}(S; \theta_t^\varepsilon), \qquad J_t^{\text{real}}(\varepsilon) := \partial_\theta \Phi_{\text{real}}(D^\varepsilon; \theta_t^\varepsilon),$$

and the *trajectory sensitivity* $u_{t,j}^\varepsilon := \frac{d}{d\varepsilon} \theta_t^\varepsilon$,

and the *explicit real-stats perturbation* $s_{t,j}^{\text{real},\varepsilon} := \frac{\partial}{\partial \varepsilon} \Phi_{\text{real}}(D^\varepsilon; \theta) \big|_{\theta=\theta_t^\varepsilon}$.

Evaluating at $\varepsilon = 0$ yields the shorthands

$$J_t^{syn} := J_t^{syn}(0), \quad J_t^{\text{real}} := J_t^{\text{real}}(0), \quad u_{t,j} := u_{t,j}^0 \tag{19}$$

$$s_{t,j}^{\text{real}} := s_{t,j}^{\text{real},0}, \quad a_t := a_t^0, \quad b_t := b_t^0. \tag{20}$$

Substituting into (16) at $\varepsilon = 0$ gives

$$\frac{d}{d\varepsilon} \mathcal{D}(a_t^\varepsilon, b_t^\varepsilon) \Big|_{\varepsilon=0} = \langle \nabla_1 \mathcal{D}(a_t, b_t), \ J_t^{syn} u_{t,j} \rangle + \langle \nabla_2 \mathcal{D}(a_t, b_t), \ s_{t,j}^{\text{real}} + J_t^{\text{real}} u_{t,j} \rangle. \tag{21}$$

**Step 4: Collect terms and sum over t.** Combining (15) and (21), we obtain the *general decomposition*

$$\mathcal{I}(z_j; S \mid S_{\text{inner}}) \;=\; \mathbb{E}_{\theta_0} \sum_{t=1}^{|T|} \Big[ \underbrace{\big\langle \nabla_2 \mathcal{D}(a_t, b_t),\, s_{t,j}^{\text{real}} \big\rangle}_{\text{explicit (real-stats)}} + \underbrace{\big\langle J_t^{\text{syn}\top} \nabla_1 \mathcal{D}(a_t, b_t) + J_t^{\text{real}\top} \nabla_2 \mathcal{D}(a_t, b_t),\, u_{t,j} \big\rangle}_{\text{implicit via trajectory}} \Big]. \tag{22}$$

**Step 5: Trajectory sensitivity $u_{t,j}$ under gradient descent (one concrete inner optimizer).**
According to Ling et al. (1984), we get:

$$\frac{d\theta^*}{d\varepsilon}\bigg|_{\varepsilon=0} \;=\; -\,H(\theta^*; D)^{-1} \nabla_\theta \ell(\theta^*; z_j), \qquad H(\theta^*; D) := \nabla_\theta^2 \ell_{\text{inner}}(\theta^*; D). \tag{23}$$

**Step 6: Special cases (choice of $S_{\text{inner}}$).**

- **Inner uses synthetic data** ($S_{\text{inner}} = S$): then $S_{\text{inner}}^\varepsilon$ is constant in $\varepsilon$ and thus $u_{t,j} \equiv 0$. Equation (22) reduces to the *outer-only* (envelope-type) term

$$\mathcal{I}(z_j; S \mid S) \;=\; \mathbb{E}_{\theta_0} \sum_{t=1}^{|T|} \big\langle \nabla_2 \mathcal{D}(a_t, b_t),\, s_{t,j}^{\text{real}} \big\rangle. \tag{24}$$

- **Inner uses real data** ($S_{\text{inner}} = D$): then $u_{t,j}$ follows the recursion (**??**) (or the stationary form (23) at convergence), and (22) includes both explicit and implicit terms:

$$\mathcal{I}(z_j; S \mid D) \;=\; \mathbb{E}_{\theta_0} \sum_{t=1}^{|T|} \Big[ \big\langle \nabla_2 \mathcal{D}(a_t, b_t),\, s_{t,j}^{\text{real}} \big\rangle \;+\; \big\langle J_t^{\text{syn}\top} \nabla_1 \mathcal{D}(a_t, b_t) + J_t^{\text{real}\top} \nabla_2 \mathcal{D}(a_t, b_t),\, u_{t,j} \big\rangle \Big]. \tag{25}$$

**Final statement.** Combining the above, the distillation influence function admits the following explicit decomposition:

$$\mathcal{I}(z_j; S \mid S_{\text{inner}}) = \mathbb{E}_{\theta_0} \sum_{t=1}^{|T|} \Big[ \underbrace{\big\langle \nabla_2 \mathcal{D}(a_t, b_t),\, s_{t,j}^{\text{real}} \big\rangle}_{\text{explicit (real-stats)}} + \underbrace{\big\langle J_t^{\text{syn}\top} \nabla_1 \mathcal{D}(a_t, b_t) + J_t^{\text{real}\top} \nabla_2 \mathcal{D}(a_t, b_t),\, u_{t,j} \big\rangle}_{\text{implicit via trajectory}} \Big], \tag{26}$$

where, at $\varepsilon = 0$,

$$a_t = \Phi_{syn}(S; \theta_t), \quad b_t = \Phi_{\text{real}}(D; \theta_t), \quad s_{t,j}^{\text{real}} = \tfrac{\partial}{\partial \varepsilon} \Phi_{\text{real}}(D^\varepsilon; \theta_t)\big|_{\varepsilon=0}, \quad u_{t,j} = \tfrac{d}{d\varepsilon} \theta_t^\varepsilon \big|_{\varepsilon=0}.$$

In particular, $u_{t,j} \equiv 0$ when $S_{\text{inner}} = S$, and $u_{t,j}$ is given by the Ling et al. (1984) (or (23) at convergence) when $S_{\text{inner}} = D$.