# OpenReview forum: "Not All Instances Are Equally Valuable: Towards Influence-Weighted Dataset Distillation"
_ICLR.cc/2026/Conference — Submitted to ICLR 2026_

### Official Review · Reviewer_iugd · 2025-10-16

**Soundness:** 2
**Presentation:** 2
**Contribution:** 2
**Rating:** 2
**Confidence:** 5

**Summary:**

The paper tackles dataset distillation under the observation that not all real samples contribute equally to synthesizing high-quality distilled data. It proposes a plug-and-play influence-weighted distillation module that assigns soft weights to real samples during the distillation process, aiming to emphasize helpful examples without discarding information. The method is positioned as complementary to multiple families of distillation algorithms.

**Strengths:**

1. The idea that certain real samples are more (or less) helpful for driving the synthetic set toward better generalization is well-framed and practically relevant.
2. The weighting mechanism is presented as a lightweight module that can be layered onto several existing distillation approaches without changing their core objectives.
3. The decomposition into explicit and implicit terms provides a semi-theoretical rationale for why weighting should influence the learning dynamics of the distilled set.

**Weaknesses:**

1. The paper does not convincingly separate itself from influential lines on data valuation/influence estimation and “prune-then-distill.” A structured comparison of objectives, assumptions, and complexity is missing.
2. The additional cost of computing (or approximating) influence scores is not reported in time/memory terms, leaving open whether the method scales to larger backbones or higher-resolution data.
3. Results are confined to low-resolution benchmarks (CIFAR/SVHN); gains concentrate in very small-data settings. It is unclear whether similar benefits hold for more realistic distributions or downstream tasks.
4. The comparison includes only methods published before 2023. Recent methods in decoupling-based distillation (e.g., SRe2L [1], RDED [2]) and diffusion-based distillation (e.g., Minimax [3], D⁴M [4]) are neither discussed nor compared.

[1] Squeeze, Recover and Relabel: Dataset Condensation at ImageNet Scale From A New Perspective

[2] On the Diversity and Realism of Distilled Dataset: An Efficient Dataset Distillation Paradigm

[3] Efficient Dataset Distillation via Minimax Diffusion

[4] D⁴M: Dataset Distillation via Disentangled Diffusion Model

**Questions:**

1. How are influence scores approximated in practice (e.g., HVPs, truncated solvers), how often are they updated, and what is the per-epoch overhead relative to each underlying distillation method?

2. Which internal signal is ultimately used when alternatives are possible, and how do the explicit vs. implicit components contribute empirically (component-wise ablation)?

3. Is there a data-scale-aware or adaptive strategy for the temperature parameter, especially across IPC and class-imbalance regimes?

4. Under a single, unified training budget, do improvements persist on stronger baselines and higher-resolution datasets (≥224px)?

5. How robust is the method to mislabeled or adversarially noisy real samples—does soft weighting amplify harmful examples under certain distributions?

---

### Official Review · Reviewer_JY8z · 2025-10-25

**Soundness:** 2
**Presentation:** 3
**Contribution:** 2
**Rating:** 4
**Confidence:** 5

**Summary:**

Given that most existing dataset distillation methods assume that all real instances contribute equally to the distillation process, this paper proposes a dataset distillation framework called Influence-Weighted Distillation (IWD). The core idea of ​​IWD is to use an influence function to estimate the contribution of each real-world data instance to the distillation objective. This method is designed as a modular plug-in that can be seamlessly integrated into various existing distillation frameworks. Experimental results show that IWD significantly improves performance on distilled datasets.

**Strengths:**

- IWD is designed as a plug-and-play module that can be combined with various distillation paradigms.
- The paper is easy to understand.

**Weaknesses:**

- The method proposed in this paper is only tested on CIFAR10, CIFAR100, and SVHN small datasets, and is not tested on ImageNet1k and ImageNet21k. Therefore, the generalization of the proposed method cannot be demonstrated.
- This paper does not measure the time and computing cost required to calculate the influence score, so it cannot effectively illustrate the cost of the proposed plug-and-play method.
- The proposed method lacks innovation, influence functions have long been used in the field of data selection. This paper simply combines it with dataset distillation.
- The influence function will have a significant impact on the estimation cost as the number of data samples in the training set increases, which makes the proposed method unsuitable for some large datasets.
- No comparison with state-of-the-art methods.
- The proposed method is not compared with existing methods [1,2] that combine core coreset selection and dataset distillation.

[1] Abbasi, Ali, et al. "Diffusion-Augmented Coreset Expansion for Scalable Dataset Distillation." arXiv preprint arXiv:2412.04668 (2024).
[2] Moser, Brian B., et al. "Distill the best, ignore the rest: Improving dataset distillation with loss-value-based pruning." arXiv preprint arXiv:2411.12115 (2024).

**Questions:**

If somebody combines some of the most advanced dataset selection methods with dataset distillation, will the effect be worse than your method?

---

### Official Review · Reviewer_jceJ · 2025-10-25

**Soundness:** 2
**Presentation:** 3
**Contribution:** 2
**Rating:** 4
**Confidence:** 4

**Summary:**

This paper proposes Influence-Weighted Distillation (IWD), a framework that improves dataset distillation by weighting real samples based on their estimated influence on the distillation objective. Unlike existing methods that treat all instances equally or prune low-quality samples, IWD leverages influence functions to assign adaptive softmax weights, emphasizing informative data while downweighting redundant or harmful ones. The method integrates seamlessly into existing bi-level distillation pipelines (e.g., DC, IDC, PDD) without structural changes. Experiments on CIFAR10, CIFAR100, and SVHN show consistent accuracy gains, demonstrating that influence-guided weighting enhances the quality and generalization of distilled datasets. Overall, IWD provides a simple yet effective and interpretable approach to account for data quality in dataset distillation.

**Strengths:**

1. The paper addresses a fundamental yet often overlooked problem in dataset distillation, that different training instances contribute unequally to the quality of the distilled dataset. The motivation is clear and intuitive, highlighting a real limitation in existing approaches.

2. The proposed method consistently improves multiple baselines (DC, IDC, PDD) across standard benchmarks (CIFAR10/100, SVHN). These results demonstrate that the influence-based weighting mechanism effectively enhances the representativeness and utility of distilled datasets.

3.  IWD is designed as a modular plug-in that can be seamlessly integrated into existing bi-level dataset distillation pipelines. Its flexibility allows broad applicability across different optimization objectives and distillation paradigms, without requiring major architectural or algorithmic modifications.

4. Beyond empirical gains, IWD provides an interpretable perspective on dataset distillation by quantifying the influence of individual training samples. This analysis clarifies which instances are beneficial, redundant, or harmful, offering meaningful diagnostic insights into data quality and model behavior.

**Weaknesses:**

1. The proposed IWD framework relies on costly influence function estimation involving Hessian–vector products and bi-level optimization, leading to heavy computation and poor scalability to large datasets like ImageNet-1k. It is also restricted to bi-level distillation frameworks (e.g., DD, DC, IDC, PDD) due to its dependence on outer-loop gradients, and cannot extend to more efficient uni-level methods (e.g., SRe2L [1], EDC [2], FADRM [3]). In contrast, Prune-then-Distill offers greater flexibility and scalability across both paradigms.

2. The cross-architecture evaluation is limited to ConvNet-3 and ResNet-10, both shallow and small models, providing weak evidence of generalization. The marginal gains under ConvNet and the absence of modern architectures such as Vision Transformers further undermine the claimed robustness. Moreover, experiments on only small-scale datasets (e.g., CIFAR10) fail to demonstrate real-world scalability. Broader studies on larger datasets and stronger backbones are needed to validate IWD’s generality.

3. Although the proposed method performs well empirically, its theoretical properties remain unexplored.

**Questions:**

1. Since the efficiency results are not explicitly presented in the table, could the authors elaborate on the time cost for dataset distillation and the corresponding GPU memory usage?

2. In Figure 5, I am confused about why certain visually distinctive samples, such as the yellow airplane with the lowest influence score, are assigned such low values. Intuitively, such unique instances seem rare and should exhibit relatively higher influence scores.

3. Additionally, the paper claims that low-influence samples are easy to recognize and redundant. Are there any quantitative experiments or analyses that support this statement?


[1] Squeeze, recover and relabel: Dataset condensation at imagenet scale from a new perspective

[2] Elucidating the Design Space of Dataset Condensation

[3] FADRM: Fast and Accurate Data Residual Matching for Dataset Distillation

---

### Official Review · Reviewer_7mmh · 2025-11-01

**Soundness:** 3
**Presentation:** 2
**Contribution:** 2
**Rating:** 2
**Confidence:** 3

**Summary:**

The paper proposes Influenced-Weighted Distillation (IWD), which leverages influence functions to provide a simple reweighing mechanism on real data that improve the performance of existing dataset distillation algorithms. The key argument is that existing dataset distillation algorithm treat every sample equally. Therefore, outlier samples may provide noisy gradients,  negatively influencing the optimization objective. The paper shows that the proposed weighting mechanism IWD improves three existing dataset distillation algorithm on CIFAR-10, CIFAR-100, and SVHN.

**Strengths:**

1. The proposed IWD algorithm is light-weight and can be plugged into any existing dataset distillation algorithm.
2. The paper provide adequate mathematical justification towards their design of the influence calculation.
3. The proposed IWD algorithm does improve three existing dataset distillation algorithm on CIFAR-10, CIFAR-100, and SVHN, with the largest on gradient matching from 44.9% to 52.7% with 10 IPC.

**Weaknesses:**

1. While the paper provide a large number of baselines for comparison, the proposed algorithm is only applied to three of the baseline: gradient matching, information-intensive dataset condensation, and progressive dataset distillation. Since the proposed method suppose to augment existing method, more comparison needs to be made. For instance, trajectory matching is the most popular dataset distillation algorithm, yet no evaluation is reported on what happens if we apply IWD on top.
2. Following 1, there is a heavy imbalance in the baseline used for evaluation. The related work presents four types of dataset distillation work: meta-model matching, gradient matching, trajectory matching, and distribution matching. Yet two of the three baselines that IWD builds on top of is gradient matching. There should be at least one baseline from each of the four types to test IWD with.
3. The baselines included are not comprehensive. It is missing the BPTT baseline [1], RaBPTT [2], RCIG [3], and newer work NCFM [3].
4. Figure 2, the figure detailing the IWD, need a caption that explains the figure.

[1] Deng, Zhiwei, and Olga Russakovsky. "Remember the past: Distilling datasets into addressable memories for neural networks." Advances in Neural Information Processing Systems 35 (2022): 34391-34404.
[2] Feng, Yunzhen, Shanmukha Ramakrishna Vedantam, and Julia Kempe. "Embarrassingly Simple Dataset Distillation." The Twelfth International Conference on Learning Representations.
[3] Loo, Noel, et al. "Dataset distillation with convexified implicit gradients." International Conference on Machine Learning. PMLR, 2023.
[4] Wang, Shaobo, et al. "Dataset distillation with neural characteristic function: A minmax perspective." Proceedings of the Computer Vision and Pattern Recognition Conference. 2025.

**Questions:**

1. How well does IWD affect the convergence of the algorithm? i.e. does reweighing reduce the number of gradient steps needed to distill the data?
2. How much extra overhead does the influence calculation add to the overall compute?
3. Does IWD improve existing dataset distillation algorithm on Tiny-ImageNet?
4. Previous work [1] has found that dataset distillation algorithms effectively captures the early training dynamics (which tend to exclude outliers [2]) but does produce distilled data where there are heterogeneous grouping in the real data influenced. Does applying IWD change this behavior?

[1] Yang, William, et al. "What is dataset distillation learning?." Proceedings of the 41st International Conference on Machine Learning. 2024.
[2] Toneva, Mariya, et al. "An Empirical Study of Example Forgetting during Deep Neural Network Learning." International Conference on Learning Representations.

---

### Meta-Review · Area_Chair_kF79 · 2026-01-04

**Summary:**

The main concerns are:

- The proposed algorithm is only applied to three of the baselines.

- There is a heavy imbalance in the baseline used for evaluation.

- It relies on costly influence function estimation and is also restricted to bi-level distillation frameworks.

- The cross-architecture evaluation is limited to ConvNet-3 and ResNet-10, both of which are shallow and small models, providing weak evidence of generalization.

- Its marginal gains under ConvNet and the absence of modern architectures such as Vision Transformers further undermine the claimed robustness.

- Theoretical properties remain unexplored.

- It has only been tested on the CIFAR10, CIFAR100, and SVHN small datasets.

- It does not measure the time and computing cost.

**Reviewer Concerns:**

The authors did not response.

**Reviewer Scores:**

Based on the above concerns, I think none of the reviewers would raise the score except that the authors further improve the full experiment section.

---

### Decision · Program_Chairs · 2026-01-26

Reject